Comparison of clinical outcomes after drug-eluting balloon and drug-eluting stent use for in-stent restenosis related acute myocardial infarction: a retrospective study

Fang Chih-Yuan
Fang Hsiu-Yu
Chen Chien-Jen
Yang Cheng-Hsu
Wu Chiung-Jen
Lee Wei-Chieh leeweichieh@yahoo.com.tw
Division of Cardiology, Department of Internal Medicine, Kaohsiung Chang Gung Memorial Hospital, Chang Gung University College of Medicine , Kaohsiung , Taiwan
Ma Shuangtao
Electronic publication date: 2018 Apr 18
Publication date: 2018
Volume: 6
Electronic Location ID: e4646
Received 2018 Jan 3; Accepted 2018 Mar 29
Copyright: ©2018 Fang et al.
Copyright year: 2018
Copyright holder: Fang et al.
License: This is an open access article distributed under the terms of the Creative Commons Attribution License, which permits unrestricted use, distribution, reproduction and adaptation in any medium and for any purpose provided that it is properly attributed. For attribution, the original author(s), title, publication source (PeerJ) and either DOI or URL of the article must be cited.
License URL: https://creativecommons.org/licenses/by/4.0/

Keywords: Drug-eluting stent, Acute myocardial infarction, Drug-eluting balloon, In-stent restenosis

Funding: The authors received no funding for this work.

==============================
Background

Good results of drug-eluting balloon (DEB) use are achieved in in-stent restenosis (ISR) lesions, small vessel disease, long lesions, and bifurcations. However, few reports exist about DEB use in acute myocardial infarction (AMI) with ISR. This study’s aim was to evaluate the efficacy of DEB for AMI with ISR.

Methods

Between November 2011 and December 2015, 117 consecutive patients experienced AMI including ST-segment elevation MI, and non-ST-segment elevation MI due to ISR, and received percutaneous coronary intervention (PCI). We divided our patients into two groups: (1) PCI with further DEB, and (2) PCI with further drug-eluting stent (DES). Clinical outcomes such as target lesion revascularization, target vessel revascularization, recurrent MI, stroke, cardiovascular mortality, and all-cause mortality were analyzed.

Results

The patients’ average age was 68.37 ± 11.41 years; 69.2% were male. A total of 75 patients were enrolled in the DEB group, and 42 patients were enrolled in the DES group. The baseline characteristics between the two groups were the same without statistical differences except for gender. Peak levels of cardiac biomarker, pre- and post-PCI cardiac function were similar between two groups. The major adverse cardiac cerebral events rate (34.0% vs. 35.7%; p = 0.688) and cardiovascular mortality rate (11.7% vs. 12.8%; p = 1.000) were similar in both groups.

Conclusions

DEB is a reasonable strategy for AMI with ISR. Compared with DES, DEB is an alternative strategy which yielded acceptable short-term outcomes and similar 1-year clinical outcomes.

Introduction

With the improvement of the technology and the design of the stent, the incidence of in-stent restenosis (ISR) decreased. Drug-eluting balloons (DEBs) have become an important alternative to the current treatment of ISR (Wöhrle et al., 2012). The 2014 European guideline for coronary revascularization recommends the use of DEB for the treatment of ISR of a bare-metal stent (BMS) or drug-eluting stent (DES) (Class I, level of evidence: B) (Windecker et al., 2014). However, the American College of Cardiology/American Heart Association/Society for Cardiovascular Intervention guidelines for percutaneous coronary intervention (PCI) recommends DES to treat BMS ISR (Class I, Level of Evidence: A) and plain old balloon angioplasty, BMS, or DES to treat DES-ISR (Levine et al., 2011). The guideline does not issue any recommendations for DEB. In real-world practice, the use of DEB for either BMS or DES restenosis showed good clinical results (Stella et al., 2011; Lee et al., 2016). DEBs provide advantages over DESs, such as rapid release of drug to the arterial wall, the absence of polymers and stent structures, and the absence of stent thrombosis (Waksman & Pakala, 2009). Paclitaxel has been identified as the primary drug for use in DEBs because of its long-lasting antiproliferative effect and retained uptake by vascular smooth muscle cells up to one week (Waksman & Pakala, 2009). On the other hand, DEB also has been applied for de novo coronary lesions, small vessel disease, long lesions, and bifurcations, and presented good results (Fröhlich et al., 2013; Vaquerizo et al., 2015; Richelsen, Overvad & Jensen, 2016).

However, there are few data about use of DEB in acute coronary syndromes, especially acute myocardial infarction (AMI). One DEB-AMI trial (Belkacemi et al., 2012) in ST-segment elevation myocardial infarction (STEMI) patients, angiographic results of DES were superior to both BMS and DEB plus BMS. However, no clinical outcomes of DEB for STEMI and no other randomized trials about DEB use in non-ST-segment elevation MI (NSTEMI) were achieved. Most trials and real-world practice also use DEB for relative stable conditions such as stable angina and unstable angina (Wöhrle et al., 2012; Lee et al., 2016; Fröhlich et al., 2013).

Therefore, we focused on the use of DEB for acute conditions with thrombus. This study’s aim was to compare the differences in clinical outcomes between the use of DES and DEB for AMI with ISR.

Materials and Methods

The study was approved by the Institutional Review Committee on Human Research of Chang Gung Memorial Hospital for retrospective analysis in consecutive patients with AMI including STEMI and NSTEMI who underwent PCI with DEB and DES for ISR between November 2011 and December 2015 in our hospital. The approval number was 201701790B0. The raw data was from the myocardial infarction registry of Kaohsiung Chang Gung Memorial Hospital.

Patients and groups

Between November 2011 and December 2015, 117 consecutive patients with AMI and received PCI for ISR were retrospectively enrolled. All patients refused CABG due to high operation risk and patients’ choice. The patients were divided into two groups: (1) PCI with further DEB, and (2) PCI with further DES. In both groups, dual anti-platelet therapy (Aspirin plus clopidogrel or ticagrelor) was used for one year for AMI. The decision of using DEB or DES was on the operator’s discretion. The most reasons that the operator choose DEB were: (1) the lesions had more than two layers of metallic stents; (2) the lesions were relatively less of a plaque burden after balloon angioplasty.

Endpoints

Clinical outcomes such as target lesion revascularization (TLR), target vessel revascularization (TVR), recurrent MI, stroke, cardiovascular mortality, and all-cause mortality were analyzed. In-hospital major adverse cardiac cerebral events (MACCEs), cardiovascular mortality, and all-cause mortality were compared between the two groups.

Definitions

AMI definitions were in accordance with the most recent universal definition of AMI (Thygesen et al., 2012). TLR was defined as a repeat PCI or CABG for a lesion in the previously treated segment or in an adjacent 5 mm segment. TVR was defined as a repeat PCI in a target vessel. MACCEs included TLR, TVR, recurrent MI, stroke, and cardiovascular mortality. Cardiovascular mortality was defined as death related to cardiac arrhythmia, heart failure, and cardiogenic shock. All-cause mortality was defined as death from any cause.

Study endpoints

The primary endpoints were a MACCE during the follow-up period. The secondary endpoint was all-cause mortality during the follow-up period.

Statistical analysis

Data are expressed as a mean ± standard deviation for continuous variables or as counts and percentages for categorical variables. Continuous variables were compared using an independent sample t or Mann–Whitney U tests. Categorical variables were compared using a Chi-square statistic. A Kaplan–Meier curve was performed with log rank test for TLR, TVR, recurrent MI, and cardiovascular mortality in DEB and DES groups during the 1-year follow-up period.

Because the patients were not randomly assigned, there was some bias in baseline characteristics. In order to compare the clinical effect between DEB and DES, a propensity score matched analysis was performed as a 1-to-1 matched analysis using a logistic regression model for the DEB group versus the DES group to adjust for differences in baseline characteristics. Using the estimated logits, the DEB group and the DES group had the closest estimated logit value. The baseline covariates were compared between these two groups and were similar.

All statistical analyses were performed using SPSS 22.0 (IBM Corp., Armonk, NY, USA). A p-value <0.05 was considered statistically significant.

Table 1 Patient characteristics of DEB and DES group.

	DEB (N = 75; L = 103)	DES (N = 42; L = 54)	P value	
General demographics				
Age (years)	67.53 ± 11.62	69.86 ± 11.00	0.292	
Male sex (%)	46 (61.3)	35 (83.3)	0.021	
Indication			0.541	
STEMI (%)	7 (9.3)	6 (14.3)		
NSTEMI (%)	68 (90.7)	36 (85.7)		
Killip classification			0.079	
I (%)	34 (45.3)	29 (69.0)		
II (%)	13 (17.3)	4 (9.5)		
III (%)	17 (22.7)	7 (16.7)		
IV (%)	11 (14.7)	2 (4.8)		
Risk factors for MI				
Diabetes (%)	58 (77.3)	26 (61.9)	0.089	
Current smoker (%)	25 (33.3)	16 (38.1)	0.687	
Hypertension (%)	64 (85.3)	37 (88.1)	0.784	
Prior MI (%)	32 (42.7)	15 (35.7)	0.556	
Prior stroke (%)	8 (10.7)	7 (16.7)	0.394	
PAOD (%)	16 (21.3)	4 (9.5)	0.129	
Dyslipidemia (%)	39 (52.0)	23 (54.8)	0.848	
CABG (%)	5 (6.7)	3 (7.1)	1.000	
ESRD on maintenance hemodialysis (%)	28 (37.3)	12 (28.6)	0.418	
Heart failure (%)	34 (45.3)	19 (45.2)	1.000	
Laboratory examination				
Creatinine (except ESRD) (mg/dL)	2.08 ± 1.09	1.46 ± 0.82	0.132	
CK-MB (ng/mL)	51.57 ± 11.84	76.59 ± 15.95	0.166	
Troponin-I (ng/mL)	18.07 ± 8.59	24.72 ± 10.86	0.244	
Left ventricular ejection fraction (%)				
Before	55.47 ± 12.06	51.42 ± 12.79	0.117	
After	59.26 ± 11.75	55.13 ± 13.08	0.182	
Characteristics of coronary artery disease				
Single or multiple-vessel disease			0.252	
Single vessel disease (%)	3 (4.0)	4 (9.5)		
Double vessel disease (%)	16 (21.3)	5 (11.9)		
Triple vessel disease (%)	56 (74.7)	33 (78.6)		
Left main disease (%)	23 (30.7)	12 (28.6)	0.837	
Previous stent			0.867	
Bare-metal stent (%)	59 (57.3)	30 (55.6)		
Drug-eluting stent (%)	44 (42.7)	24 (44.4)		
Infarcted artery			0.174	
Left main (%)	1 (1.0)	3 (5.6)		
Left anterior descending artery (%)	46 (44.7)	21 (38.9)		
Left circumflex artery (%)	25 (24.3)	9 (16.7)		
Right coronary artery (%)	31 (30.1)	21 (38.9)		
One-year follow-up angiography (%)	34 (45.3)	23 (54.8)	0.343	
Notes.

Data are expressed as mean ± SD or as number (percentage).

N number

L lesion

DEB drug-eluting balloon

DES drug-eluting stent

STEMI ST-segment elevation myocardial infarction

NSTEMI non ST-segment elevation myocardial infarction

MI myocardial infarction

PAOD peripheral arterial occlusive disease

CABG coronary artery bypass grafting

ESRD end stage renal disease

CK-MB creatine kinase-MB

Results

Patient characteristics (Table 1)

The average age of the patients in both groups was similar, but the percentage of males was lower in the DEB group (61.3% vs. 83.3%; p = 0.021). Most patients presented NSTEMI in the DEB and DES groups (90.7% vs. 83.3%), and most patients presented Killip I status. The risk factors for AMI were similar between the two groups. The level of serum creatinine was similar in the both groups (2.08 ± 1.09 mg/dL vs. 1.46 ± 0.82 mg/dL; p = 0.132). Similar prevalence of multiple vessel coronary artery disease (96.0% vs. 90.5%) and left main coronary artery disease (30.7% vs. 28.6%) was observed in both groups. The previous stents and the infarcted artery were similar between groups. The prevalence of follow-up coronary angiography was 45.3% in the DEB group and 54.8% in the DES group (p = 0.343).

Lesion characteristics (Table 2)

Most lesions were diffuse (63.1% vs. 59.3%) in both groups. Greater pre-PCI stenotic percentage (82.84 ± 12.33% vs. 78.55 ± 8.99%; p = 0.025) and greater post-PCI stenotic percentage (16.68 ± 8.88% vs. 12.03 ± 6.10%; p = 0.001) were seen in the DEB group. Lower post-PCI minimal luminal diameter (MLD) (2.50 ± 0.55 mm vs. 2.89 ± 0.56 mm; p < 0.001) and post-PCI reference luminal diameter (RLD) (3.05 ± 0.65 mm vs. 3.29 ± 0.60 mm; p = 0.028) were seen in the DEB group. The percentage of intravascular ultrasound study (IVUS) use was similar in both groups (32.0% vs. 38.9%; p = 0.480). Smaller diameter of the size of DEB (3.08 ± 0.42 mm vs. 3.23 ± 0.43 mm; p = 0.042) and similar length of the DEB (26.41 ± 4.20 mm vs. 28.02 ± 7.81 mm; p = 0.095) were noted when comparing with the size of DES.

Table 2 Lesion characteristics of DEB and DES group.

	DEB (N = 75; L = 103)	DES (N = 42; L = 54)	P value	
Lesion type			0.730	
Focal lesion (%)	38 (36.9)	22 (40.7)		
Diffuse lesion (%)	65 (63.1)	32 (59.3)		
The characteristics of lesion				
Pre-PCI stenosis (%)	82.84 ± 12.33	78.55 ± 8.99	0.025	
MLD (mm)	0.52 ± 0.40	0.60 ± 0.30	0.185	
RLD (mm)	2.93 ± 0.65	2.88 ± 0.63	0.668	
Post-PCI stenosis (%)	16.68 ± 8.88	12.03 ± 6.10	0.001	
MLD (mm)	2.50 ± 0.55	2.89 ± 0.56	<0.001	
RLD (mm)	3.05 ± 0.65	3.29 ± 0.60	0.028	
The use of intravascular ultrasound study (%)	33 (32.0)	21 (38.9)	0.480	
The characteristics of DEB or DES				
Diameter (mm)	3.08 ± 0.42	3.23 ± 0.43	0.042	
Length (mm)	26.41 ± 4.20	28.02 ± 7.81	0.095	
Notes.

Data are expressed as mean ± SD or as number (percentage).

N number

L lesion

DEB drug-eluting balloon

DES drug-eluting stent

PCI percutaneous coronary intervention

MLD minimal luminal diameter

RLD reference luminal diameter

One-year clinical outcomes of DEB and DES group (Table 3)

Between two groups, the incidence of events including in-hospital MACCE, total MACCE, TLR, TVR, recurrent MI, stroke, cardiovascular death, and all-cause mortality were similar.

Table 3 One-year clinical outcomes of DEB and DES group.

	DEB (N = 75; L = 103)	DES (N = 42; L = 54)	P value	
In-hospital MACCE (%)	2 (2.7)	3 (7.1)	0.348	
MACCE (%)	24 (34.0)	15 (35.7)	0.688	
Target-lesion revascularization (%)	21 (26.9)	9 (22.5)	0.661	
Target-vessel revascularization (%)	25 (31.3)	11 (27.5)	0.833	
Recurrent myocardial infarction (%)	9 (16.1)	7 (21.9)	0.570	
STEMI (%)	0	1 (14.3)		
NSTEMI (%)	9 (100)	6 (85.7)		
Stroke (%)	2 (3.6)	1 (3.1)	1.000	
Cardiovascular mortality (%)	7 (11.7)	5 (12.8)	1.000	
All-cause mortality (%)	16 (22.9)	7 (17.1)	0.628	
Notes.

Data are expressed as mean ± SD or as number (percentage).

N number

L lesion

DEB drug-eluting balloon

DES drug-eluting stent

MACCE major adverse cardiac cerebral event

STEMI ST-segment elevation myocardial infarction

NSTEMI non ST-segment elevation myocardial infarction

The Kaplan–Meier curves of 1-year clinical outcomes of DEB and DES groups in TLR, TVR, recurrent MI, and cardiovascular mortality (Fig. 1)

In Fig. 1A, the Kaplan–Meier curve of 1-year TLR showed significant difference at the half-year follow-up period (6.3% vs. 20.9%; p = 0.034), and became no different at the 1-year follow-up period (26.9% vs. 22.5%; p = 0.862). In Fig. 1B, the Kaplan–Meier curve of 1-year TVR showed no significant difference at the half-year follow-up period (11.1% vs. 23.3%; p = 0.114), and at 1-year follow-up period (31.3% vs. 27.5%; p = 0.776). In Fig. 1C, the Kaplan–Meier curve of recurrent MI showed no significant difference at the half-year follow-up period (5.1% vs. 14.7%; p = 0.136), and at the 1-year follow-up period (16.1% vs. 21.9%; p = 0.400). In Fig. 1D, the Kaplan–Meier curve of cardiovascular death showed no significant difference at the half-year follow-up period (6.5% vs. 12.5%; p = 0.309), and at the 1-year follow-up period (11.7% vs. 12.8%; p = 0.765).

Figure 1 The Kaplan–Meier curves of 1-year clinical outcomes of DEB and DES group in TLR, TVR, recurrent MI, and cardiovascular mortality.

(A) The Kaplan–Meier curve of 1-year TLR: significant difference at the half-year follow-up period (6.3% vs. 20.9%; p = 0.034) was noted. No difference at 1-year follow-up period (26.9% vs. 22.5%; p = 0.862) was noted. (B) The Kaplan-Meier curve of 1-year TVR: no significant difference was noted at the half-year follow-up period (11.1% vs. 23.3%; p = 0.114), and at the 1-year follow-up period (31.3% vs. 27.5%; p = 0.776). (C) The Kaplan-Meier curve of 1-year recurrent myocardial infarction: no significant difference was noted at at the half-year follow-up period (5.1% vs. 14.7%; p = 0.136), and at 1-year follow-up period (16.1% vs. 21.9%; p = 0.400). (D) The Kaplan-Meier curve of 1-year cardiovascular mortality: The Kaplan-Meier curve of 1-year TVR: no significant difference was noted at the half-year follow-up period (6.5% vs. 12.5%; p = 0.309), and at 1-year follow-up period (11.7% vs. 12.8%; p = 0.765).

Patient and lesion characteristics, and one-year clinical outcomes of DEB and DES group after propensity score matched (Tables 4 and 5)

After propensity score matched, all baseline characteristic became similar between two groups except post-PCI MLD. In addition, higher incidence of TLR (4.9% vs. 22.0%; p = 0.048) and TVR (4.9% vs. 24.9%; p = 0.026) was noted at the half-year follow-up period.

Table 4 Patient and lesion characteristics of DEB and DES group after propensity score matched.

	DEB (N = 40; L = 52)	DES (N = 40; L = 52)	P value	
General demographics				
Age (years)	67.78 ± 13.04	69.48 ± 11.00	0.530	
Male sex (%)	34 (85.0)	33 (82.5)	1.000	
Indication		0.481		
STEMI (%)	3 (7.5)	6 (15.0)		
NSTEMI (%)	37 (92.5)	34 (85.0)		
Killip classification				
≥III (%)	10 (25.0)	9 (22.5)	1.000	
Risk factors for MI				
Diabetes (%)	24 (60.0)	26 (65.0)	0.818	
Current smoker (%)	20 (50.0)	15 (37.5)	0.367	
Hypertension (%)	35 (87.5)	35 (87.5)	1.000	
Prior MI (%)	15 (37.5)	13 (32.5)	0.815	
Prior stroke (%)	5 (12.5)	7 (17.5)	0.755	
PAOD (%)	6 (15.0)	4 (10.0)	0.737	
Dyslipidemia (%)	24 (60.0)	23 (57.5)	1.000	
CABG (%)	4 (10.0)	3 (7.5)	1.000	
ESRD on maintenance hemodialysis (%)	13 (32.5)	12 (30.0)	1.000	
Heart failure (%)	14 (30.0)	18 (45.0)	0.494	
Laboratory examination				
CK-MB (ng/mL)	58.42 ± 17.24	68.86 ± 19.71	0.638	
Troponin-I (ng/mL)	17.92 ± 8.91	22.45 ± 9.75	0.492	
Left ventricular ejection fraction (%)				
Before	53.59 ± 11.96	50.95 ± 13.93	0.059	
After	58.94 ± 11.84	55.72 ± 13.19	0.137	
Characteristics of coronary artery disease				
Multiple-vessel disease	30 (75.0)	32 (80.0)	0.620	
Left main disease (%)	13 (32.5)	12 (30.0)	1.000	
Previous stent			1.000	
Bare-metal stent (%)	30 (57.7)	30 (57.7)		
Drug-eluting stent (%)	22 (42.3)	22 (42.3)		
Infarcted artery			0.330	
Left main (%)	0 (0)	3 (5.8)		
Left anterior descending artery (%)	20 (38.5)	20 (38.5)		
Left circumflex artery (%)	12 (23.1)	9 (17.3)		
Right coronary artery (%)	20 (38.5)	20 (38.5)		
Lesion type			0.692	
Focal lesion (%)	24 (46.2)	21 (40.4)		
Diffuse lesion (%)	28 (53.8)	31 (59.6)		
The characteristics of lesion				
Pre-PCI stenosis (%)	81.50 ± 12.75	78.65 ± 9.11	0.193	
MLD (mm)	0.59 ± 0.43	0.59 ± 0.30	0.975	
RLD (mm)	2.95 ± 0.60	2.88 ± 0.64	0.553	
Post-PCI stenosis (%)	14.83 ± 8.04	12.07 ± 6.19	0.052	
MLD (mm)	2.64 ± 0.59	2.89 ± 0.57	0.032	
RLD (mm)	3.10 ± 0.69	3.29 ± 0.61	0.140	
The use of intravascular ultrasound study (%)	13 (32.5)	15 (37.5)	0.815	
The characteristics of DEB or DES				
Diameter (mm)	3.16 ± 0.43	3.23 ± 0.44	0.462	
Length (mm)	26.50 ± 4.21	27.64 ± 7.70	0.353	
Notes.

Data are expressed as mean ± SD or as number (percentage).

N number

L lesion

DEB drug-eluting balloon

DES drug-eluting stent

STEMI ST-segment elevation myocardial infarction

NSTEMI non ST-segment elevation myocardial infarction

MI myocardial infarction

PAOD peripheral arterial occlusive disease

CABG coronary artery bypass grafting

ESRD end stage renal disease

CK-MB creatine kinase-MB

PCI percutaneous coronary intervention

MLD minimal luminal diameter

RLD reference luminal diameter

Table 5 One-year clinical outcomes of DEB and DES group after propensity score matched.

	DEB (N = 40; L = 52)	DES (N = 40; L = 52)	P value	
In-hospital MACCE (%)	0 (0)	3 (7.5)	0.241	
Half-year				
Target-lesion revascularization (%)	2 (4.9)	9 (22.0)	0.048	
Target-vessel revascularization (%)	2 (4.9)	10 (24.4)	0.026	
One-year				
MACCE (%)	11 (27.5)	15 (37.5)	0.474	
Target-lesion revascularization (%)	14 (34.1)	9 (23.7)	0.333	
Target-vessel revascularization (%)	14 (34.1)	11 (28.9)	0.638	
Recurrent myocardial infarction (%)	3 (10.0)	7 (23.3)	0.299	
Stroke (%)	1 (3.2)	1 (3.3)	1.000	
Cardiovascular mortality (%)	0 (0)	5 (13.5)	0.060	
All-cause mortality (%)	6 (16.7)	7 (17.9)	1.000	
Notes.

Data are expressed as mean ± SD or as number (percentage).

N number

L lesion

DEB drug-eluting balloon

DES drug-eluting stent

MACCE major adverse cardiac cerebral event

Discussion

In the present study, the baseline characteristics were similar, expect for gender. Most patients had multiple vessel coronary artery disease and had undergone BMS implantation previously. Most ISR were diffuse lesions in both groups. DEB group had worse pre-PCI stenotic percentage and worse post-PCI results than the DES group. These results were related to no metallic structure in the DEB group, but did not influence clinical outcomes. The DEB group experienced less recurrent MI during the half-year and 1-year follow-up periods due to no metallic structure. In the DEB group, better results at about half-year TLR and half-year TVR were noted, but similar results at about 1-year TLR and TVR were noted. A relatively lower percentage of IVUS use was related to an emergent condition. According to our results, use of DEB for ISR with AMI had similar results as use of DES, and could decrease the possibilities of short-term events.

Paclitaxel is the most effective drug used with DEB technology due to its significant lipophilia, which allows for a more homogeneous distribution through the vessel wall, as well as a quick absorption and the duration of the effect, which may be extended for several days (Waksman & Pakala, 2009). Many studies including randomized controlled trials and meta-analyses showed good results of DEB for ISR compared with conventional balloon angioplasty (Habara et al., 2011; Indermuehle et al., 2013), and similar results of DEB for ISR compared with DES (Alfonso et al., 2014b). Most patients presented a relatively stable condition such as silent ischemia, stable angina, and unstable angina in randomized controlled trials (Habara et al., 2011; Indermuehle et al., 2013; Alfonso et al., 2014b). However, there are few data about DEB in acute coronary syndromes, especially AMI. Compared with BMS alone for NSTEMI, patients treated with BMS plus DEB had significantly less luminal loss, but the treatment did not affect patient clinical outcomes (Besic et al., 2015). In STEMI, angiographic results of DES were superior to both BMS and DEB plus BMS (Belkacemi et al., 2012). Therefore, physicians preferred to use the metallic stent first for AMI due to high thrombus condition, and used the DEB to mimic DES. Currently, no head-to-head randomized study has been performed to compare the clinical outcome between only DEB and DES for AMI.

DES improved the outcome of AMI after PCI, but recurrent ISR and stent thrombosis still were difficult problems. “Stent in stent” treatment increases the possibilities of stent thrombosis due to luminal loss, chronic inflammation and hypersensitivity reactions (Alfonso et al., 2014a). Repeat stenting for ISR may be related to insufficient stent expansion and suboptimal stent geometry because restenotic or thrombosed stents are difficult to reopen (Seedial et al., 2013). DEBs have recently had a potential to overcome the limitations of DESs. Some limitations of DESs are the need for long lengths to cover the entire surface of a diseased vessel, their association with excessive intimal hyperplasia, and difficulties about adaptive remodeling of restenosis (Seedial et al., 2013). In the present study, short-term data also showed higher incidence of TLR, TVR, and recurrent MI in the DES group. Therefore, DEBs have emerged as a potential alternative to the current treatment of ISR, and provide the freedom of polymers and stent structures. Therefore, DEBs prevent the problem of “stent in stent”, do not cause stent thrombosis and luminal loss, and may decrease the possibility of sudden death if acute stent thrombosis happens after repeat stenting. In addition, few have reported about DEB use for combined ISR and acute coronary syndrome, especially AMI with high thrombus contained condition.

In the present study, the patients in both groups had similar baseline characteristics, even though non-randomized controlled study. Use of DEBs seems to provide good short-term outcomes and less TLR, TVR, and recurrent MI due to no stent thrombosis. However, the clinical outcome became similar at 1-year follow-up period. In both groups, a relatively higher event rate was noted, because the study population experienced recurrent ISR and had multiple comorbidities including diabetes, and ESRD, and multiple vessel coronary artery disease. CABG could be an option for the patient experienced recurrent ISR and had multiple vessel coronary artery disease. In this study, all patients refused CABG due to high operation risk and patients’ choice. In this study, we focused on the impact of DEB for the combination of AMI and ISR in clinical practice for a high risk population.

Limitations

The present study had some limitations, including being a non-randomized study and having selection bias because the operator may consider the use of DES in the patients with complex lesions and the use of DEB in the patients with shorter and simple lesions. Even though the present study is non-randomized, the baseline characteristics were very similar between the two groups. In addition, no previous study compared clinical outcomes between only DEB and DES use for AMI with ISR. Our study provided the insight on the use of DEB for AMI in clinical practice for high risk population.

Conclusions

DEB is a reasonable strategy for AMI with ISR. Compared with DES, DEB was an alternative strategy which yielded acceptable short-term outcomes and similar 1-year clinical outcomes.

Supplemental Information

Data S1 DEB for MI raw data

Click here for additional data file.

Additional Information and Declarations

Competing Interests

Author Contributions

Human Ethics

Data Availability

The authors declare there are no competing interests.

Chih-Yuan Fang conceived and designed the experiments, performed the experiments, prepared figures and/or tables, approved the final draft.

Hsiu-Yu Fang performed the experiments, prepared figures and/or tables, approved the final draft.

Chien-Jen Chen, Cheng-Hsu Yang and Chiung-Jen Wu performed the experiments, approved the final draft.

Wei-Chieh Lee conceived and designed the experiments, performed the experiments, analyzed the data, contributed reagents/materials/analysis tools, authored or reviewed drafts of the paper, approved the final draft.

The following information was supplied relating to ethical approvals (i.e., approving body and any reference numbers):

The study was approved by the Institutional Review Committee on Human Research of Chang Gung Memorial Hospital.

The following information was supplied regarding data availability:

The raw data are provided in Data S1.

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
