# Peer review of "Comparison of clinical outcomes after drug-eluting balloon and drug-eluting stent use for in-stent restenosis related acute myocardial infarction: a retrospective study"

_PeerJ, doi:10.7717/peerj.4646_

## Round 0.1 · original submission · Minor Revisions

The language needs to be improved . Please also address all the concerns raised by the reviewers.

·

Basic reporting

The MS should be checked carefully and some grammar errors need to be corrected, eg. line 73-74, Compared with DES, DEB__an alternative strategy …

Experimental design

No comment.

Validity of the findings

Does the gender difference account for the results?

Additional comments

Please describe the patients' classification in detail. Were the patients divided randomly?

Reviewer 2 ·

Basic reporting

Major comments:
1. The language of the manuscript should be polished, improve the expression and reorganize some of the paragraphs. For example, “however” in line 116-119, 223-225, should be used carefully.

Minor points:
1. Format all the literature references, the citation should be inside the hallmarks;
2. Acute MI could be shortened as AMI;
3. Figure part, not well labeled and described. Figure 1C legend, line 373, two “at”; Remove the label of "Percent survival" in Figure 1A, 1B and 1C. Remove number at risk in Figure 1.

Experimental design

In this study, Fang et al get the conclusion that DEB is a reasonable strategy for AMI based on a non-randomize single center clinical study by comparing the short-term outcomes with DES. Especially, the average conditions of the AMI patients are worse in DEB group, the 1 year outcome indicators are better than the DES group. The project is meaningful due to high risk of ISR after DES treatment and limited direct evidence of advantages of DEB so far.

Validity of the findings

no comment

Additional comments

no comment

·

Basic reporting

This is a well-written manuscript that reports the finding of sound and an interesting study revealing new information where the authors have evaluated the efficacy of drug-eluting balloon for acute Myocardial Infarction (MI) with in-stent restenosis (ISR) lesions for the treatment of acute myocardial infarction. There are different guidelines and recommendations for the use of DEB in Europe and America which leaves the surgeons ambiguous about the treatment regimen to be followed. In dearth of studies on the usage of DEB in acute myocardial infarction, the results presented in this study would be helpful to come to a validated conclusion. The authors have compared the differences in clinical outcome between the use of DES and DEB for acute myocardial Infarction with ISR and showed that DEB had good clinical results. The writing quality, clarity of thoughts and knowledge of the subject under study has been demonstrated. The study is presented well and the results are properly interpreted and correlated with the published literature had no major scientific issues, so I straight away forwarded it for acceptance with minor corrections in English at few places.

Experimental design

The experimental designs are acceptable with a considerable number of patients enrolled in DEB and DES groups.

Validity of the findings

The authors have done a robust study analyzing the wide number of sample and over a long period of time. The clinical outcomes chosen including target lesion revascularization, target vessel revascularization, resurrect MI, cardiovascular mortality etc. are taken into consideration and well supported by statistical analysis.

Reviewer 4 ·

Basic reporting

Fang et al conducted a retrospective analysis to compare the clinical outcomes of DEB and DES in treatment of acute myocardial infraction due to ISR. The authors reported that DEB treatment alone had similar clinical outcomes with DES alone after one-year follow up. The use of DEB is attractive in clinical setting as it potentially reduces the late-developed stent malapposition and stent thrombosis. The question raised in this study is important and the results are interesting.

Experimental design

This is a retrospective study and the design is sound.

Validity of the findings

The study is limited in this area and the data can support the main conclusion.

Additional comments

Fang et al conducted a retrospective analysis to compare the clinical outcomes of DEB and DES in treatment of acute myocardial infraction due to ISR. The authors reported that DEB treatment alone had similar clinical outcomes with DES alone after one-year follow up. The use of DEB is attractive in clinical setting as it potentially reduces the late-developed stent malapposition and stent thrombosis. The question raised in this study is important and the results are interesting. However there are some major concerns:
1) Currently, It is unclear the role of DEB alone in treating patients with acute myocardial infraction. During this study period, 64% (75/117) patients received DEB treatment while only 36% (42/117) patients received DES treatments. DEB use is pretty high given that this is a retrospective analysis. What criteria did physician use in clinic to choose the DEB or DES to treat these patients? How could physicians ensure these patients benefit most from the treatments as the basic characteristics are similar between DEB and DES group?
2) The Laboratory examination is limited, with only creatinine was listed in the Table 1. The authors need to show and compare more biomarkers, especially those related with cardiac injury and function, such as TNT, BNP
3) Although there is no difference between DEB and DES in terms of clinical outcome during a short -term follow up, both post-PCI minimal luminal diameter and post-PCI reference diameter are significantly lower in DEB group than those in DES group. How do the authors explain it? Does DEB worsen other cardiac function? Did these patients receive echocardiography before and after PCI? This is important and needs to be analysis more. These results should also be described in Abstract.
4) The language is poor and confusing throughout. It needs to be reviewed and edited by a native English speaker.

---

## Round 0.2 · accepted · Accept

All concerns have been addressed.

·

Basic reporting

no comment

Experimental design

no comment

Validity of the findings

no comment

Additional comments

This manuscript has been revised and answered the concern.

Reviewer 4 ·

Basic reporting

The authors have addressed my questions. I recommend acceptance of the manuscript.

Experimental design

The authors have addressed my questions. I recommend acceptance of the manuscript.

Validity of the findings

The authors have addressed my questions. I recommend acceptance of the manuscript.